# The Effect of Aflatoxin B1 on Tumor-Related Genes and Phenotypic Characters of MCF7 and MCF10A Cells

**DOI:** 10.3390/ijms231911856

**Published:** 2022-10-06

**Authors:** Mowaffaq Adam Ahmed Adam, Laina Zarisa Muhd Kamal, Mahibub Kanakal, Dinesh Babu, Saad Sabbar Dahham, Yasser Tabana, Bronwyn Lok, Brittany M. Bermoy, Muhammad Amir Yunus, Leslie Thian Lung Than, Khaled Barakat, Doblin Sandai

**Affiliations:** 1Department of Biomedical Science, Advanced Medical and Dental Institute, Universiti Sains Malaysia, Penang 13200, Malaysia; 2Department of Chemistry and Biochemistry, San Diego State University, San Diego, CA 92182, USA; 3Faculty of Pharmacy, University College MAIWP International, Kuala Lumpur 68100, Malaysia; 4Faculty of Pharmacy and Pharmaceutical Sciences, University of Alberta, Edmonton, AB T6G 2E1, Canada; 5Department of Science, University of Technology and Applied Sciences Rustaq, Rustaq 10 P.C. 329, Oman; 6Department of Medical Microbiology and Parasitology, Faculty of Medicine and Health Sciences, Universiti Putra Malaysia, Serdang 43400, Malaysia

**Keywords:** aflatoxin B1, MCF7, MCF10A, carcinogen, breast cancer

## Abstract

The fungal toxin aflatoxin B1 (AB1) and its reactive intermediate, aflatoxin B1-8, 9 epoxide, could cause liver cancer by inducing DNA adducts. AB1 exposure can induce changes in the expression of several cancer-related genes. In this study, the effect of AB1 exposure on breast cancer MCF7 and normal breast MCF10A cell lines at the phenotypic and epigenetic levels was investigated to evaluate its potential in increasing the risk of breast cancer development. We hypothesized that, even at low concentrations, AB1 can cause changes in the expression of important genes involved in four pathways, i.e., p53, cancer, cell cycle, and apoptosis. The transcriptomic levels of *BRCA1*, *BRCA2*, *p53*, *HER1*, *HER2*, *cMyc*, *BCL2*, *MCL1*, *CCND1*, *WNT3A*, *MAPK1*, *MAPK3*, *DAPK1*, *Casp8*, and *Casp9* were determined in MCF7 and MCF10A cells. Our results illustrate that treating both cells with AB1 induced cytotoxicity and apoptosis with reduction in cell viability in a concentration-dependent manner. Additionally, AB1 reduced reactive oxygen species levels. Phenotypically, AB1 caused cell-cycle arrest at G1, hypertrophy, and increased cell migration rates. There were changes in the expression levels of several tumor-related genes, which are known to contribute to activating cancer pathways. The effects of AB1 on the phenotype and epigenetics of both MCF7 and MCF10A cells associated with cancer development observed in this study suggest that AB1 is a potential risk factor for developing breast cancer.

## 1. Introduction

Aflatoxin B1 (AB1) is a mycotoxin produced by *Aspergillus flavus* and *Aspergillus parasiticus*, which grows on different food commodities under high temperatures and humid conditions [1]. Humans and animals can consume this carcinogenic toxin through contaminated food and water [2]. Some of the food susceptible to contamination by *Aspergillus* species, subsequently with AB1, include seeds, cereals, spices, tree nuts, and milk. Additionally, AB1 contamination was reported in tobacco leaves, dry soil, and groundwater after the rainy season [3,4,5,6,7,8]. AB1 is liposoluble, absorbed easily at the site of exposure, such as the mucus membrane in the nose, mouth, and respiratory and gastrointestinal tracts, from where it can be circulated to other parts of the body through the bloodstream [2]. Due to its chemical and physical properties with its ability to penetrate cell membranes, attach to DNA, and cause irreversible mutation, the carcinogenic effect of AB1 is ranked the highest among all the mycotoxins [1]. Additionally, AB1 was found to accumulate in the tissue of animals [9] and humans [10]. Once inside the cell, the stable form of AB1 is metabolized by the enzyme cytochrome P450, resulting in the production of highly unstable aflatoxin-8, 9-epoxide. In this unstable form, the toxin binds to DNA molecules to gain stability at high affinity causing the formation of aflatoxin-N7-guanine [11]. At this point, a transversion mutation takes place where a guanine (G) base will be transversed to a thymine (T) base [12]. This mutation was proven to have a direct effect on the cell cycle and *p53* gene, promoting tumorigenicity and cancer [12]. *p53* is an important tumor-suppressing gene and its mutation was reported in almost all types of cancer, with the highest incidence in breast and ovarian cancer [13,14]. Other important tumor-suppressing genes that, once mutated, result in the occurrence of breast cancer include *BRCA1* and *BRCA2* [15,16]. Since AB1 has a mutagenesis effect on *p53* [17], we asked several questions: Can AB1 be another oncogene driving breast cancer? Can AB1 act as an epigenetic switch controlling various pathways leading to the development of cancer? Finally, can AB1 be added to the several proven risk factors contributing to the development of breast cancer? AB1 has the ability to drive liver cancer via transversion mutation. Guanine is substituted by thymine bases in the *p53* gene, inducing epigenetic alteration and transcriptional silencing in this important tumor-suppressing gene. Consequently, this leads to kidney cancer [18], but this has never been proven in breast cancer until today. In this study, we hypothesized that AB1 exposure will induce variable changes in the expression of several genes associated with different epigenetic mechanisms. These changes will result in altering selected oncogenes, tumor-suppressing genes, cell-cycle genes, and apoptosis genes. In this study, we intended to evaluate the carcinogenic effect of AB1 on tumor-suppressing genes (*BRCA1*, *BRCA2*, and *p53*), oncogenes (*HER1*, *HER2*, and *cMyc*), cell cycle genes (*CCND1*, *WNT3A MAPK1*, and *MAPK3*), and apoptosis genes (*BCL2*, *MCL1*, *DAPK1*, *Casp8*, and *Casp9*). Each of these groups of genes represents an important part of pathways contributing to cancer progression. Once these genes are upregulated or downregulated due to the presence of an oncometabolite, different cancer types will develop. In addition, previous work has indicated that these gene groups are involved in tuning cancer phenotype. So, in this study, the selected genes were chosen due to their extensive backgrounds in previous studies. We acknowledged that several other genes are involved in these pathways, including *PALB2*, *RAD51*, and *CHEK2* [2,3] have also been implicated in cancer; however, they were not selected for this study. Additional genes from these pathways may be the subject of future studies in connection to AB1. To test our hypothesis, the expression levels of specific genes related to cancer development were quantified in the two most frequently studied breast cell lines in the literature, the human breast cancer cell line MCF7 and human normal breast cell line MCF10A. Moreover, the phenotypical changes associated with the changes in gene expression were also studied. The results of this study will give us an insight into the potential effects of this toxin in increasing the risk of breast cancer.

## 2. Results

### 2.1. AB1-Induced Cytotoxicity and Apoptotic Cell Death with Reduction in Viability of MCF7 and MCF10A Cell Lines

The toxic effects of AB1 on MCF7 cells were examined, and AB1 decreased the cell viability of MCF7 cells in a concentration-dependent manner (Figure 1A) as assessed by a cell proliferation kit post-treatment at different concentrations (1.2, 2.4, 4.8, and 9.6 µg/mL) for 48 h. The cell viabilities of the MCF7 cells were 91.7% at 1.2 µg/mL, 80% at 2.4 µg/mL, 55.78% at 4.8 µg/mL, and 31.13% at 9.6 µg/mL, indicating the toxic effect at higher concentrations. When MCF10A cells were treated with AB1, as demonstrated in Figure 1B, the control cells showed 100% viability, 98% at 2.3 µg/mL, 78% at 4.6 µg/mL, 54.47% at 9.2 µg/mL, and 20.26% 18.4 µg/mL. The evaluation of the modes of cell death in MCF7 (Figure 1C) and MCF10A (Figure 1D) cells with AB1 was carried out using annexin V-FITC apoptosis detection kit by flow cytometry analysis after 48-h treatment with different concentrations of AB1 (1.2, 2.4, 4.8, and 9.6 µg/mL). The control MCF7 cell population without AB1 treatment showed 98% viable cells with only 1% apoptotic and 0.5% necrotic cell population. MCF7 cells treated with 1.2 µg/mL AB1 showed 3.6% apoptotic, 0.1% necrotic, and 96.2% viable cells. Treatment with 2.4 µg/mL of AB1 showed 6.7% apoptotic, 0.1% necrotic, and 93.2% viable cells. Treatment with 4.8 µg/mL induced 18.4% apoptotic and 6.1% necrotic dead cells and 75.5% viable cells. Lastly, treatment with 9.6 µg/mL AB1 resulted in 46.44%, 0.95%, and 53.61% of apoptotic, necrotic, and viable cells, respectively (Figure 1C). In the case of the MCF10A cells, the control cells showed 0.8% apoptotic, 0.4 necrotic and 98.8% living cells. Treatment of 2.3 µg/mL AB1 resulted in 1.5% apoptotic, 0.5 necrotic, and 98% living cells. At a concentration of 4.6 µg/mL AB1 treatment, 1.5% apoptotic, 1.5% necrotic, and 95% viable cells were observed. At 9.2 µg/mL AB1, 32.9% apoptosis, 9.86% necrosis and 57.24% viable cell. The highest tested concentration of 18.4 µg/mL AB1 resulted in 40.84%, 7.68%, and 51.49%, apoptotic, necrotic, and viable cells, respectively (Figure 1D). These results indicated that treatment of AB1 induced concentration-dependent apoptotic cell death in breast cancer cells.

### 2.2. MCF7 and MCF10A Cells Showed a Slight Decrease in Reactive Oxygen Species Levels Post Treatment of AB1

The effect of AB1 on the intracellular reactive oxygen species (ROS) production on MCF7 cells (at final concentrations of 1.2, 2.4, 4.8, and 9.6 µg/mL) and on MCF10A cells (at final concentrations of 2.3, 4.6, 9.2, and 18.4 µg/mL) was examined using an ROS Detection Cell-Based Assay Kit (DHE). In the case of the MCF7 cells, AB1 at a concentration of 1.2 µg/mL reduced the intracellular ROS level to 55.49 ± 2.4% when detected by DHE (Figure 2A). When the MCF7 cells were treated with 2.4 and 4.8 µg/mL, the ROS levels recorded were 92.4 ± 1.2% and 91.8 ± 0.2%, respectively. Finally, treatment of MCF7 cells with 9.6 µg/mL of AB1 showed an ROS level of 85.9 ± 1%, while, on the other hand, the effect of AB1 on the intracellular ROS levels of the MCF10A cells showed an 86.8%, 93.5%, 84.1%, and 43.4% drop in ROS levels at 2.3, 4.6, 9.12, and 18.4 µg/mL AB1 as demonstrated in Figure 2B.

### 2.3. AB1 Caused Hypertrophy and Cell-Cycle Arrest but Increased Cell Migration Rates in MCF7 and MCF10A cells

The effects of AB1 at final concentrations of 1.2 and 2.3 µg/mL were investigated for any changes in morphology (Figure 3B) and cell size (Figure 4A,B) of MCF7 and MCF10A cells post treatment of AB1 for 24, 48, and 72 h were carried out using an inverted light microscope (OLYMPUS). Both the cells showed mid-morphological changes at 48 h with characteristic signs of blebbing at 72 h following AB1 treatment (Figure 3, last row). When the MCF7 cells were treated with AB1 at the three time points, the average cell size was 285 µm^2^ at 0 h, 260 µm^2^ at 24 h, 250 µm^2^ at 48 h, and 360 µm^2^ at 72 h, as demonstrated in Figure 4A (* *p* ˂ 0.05). These results demonstrated that AB1 at a final concentration of 1.2 µg/mL caused a change in MCF7 cell size where cells continued to reduce in size from 0 h to 24 h, decreased further at 48 h, and reached a maximum at 72 h. When the MCF10A cells were treated with AB1, no change in cell morphology took place at all the time points. The average cell size of MCF10A cells, when treated with AB1, showed an increase throughout the three time points and the average cell size was 55 µm^2^ at 0 h, 105 µm^2^ at 24, 102 µm^2^ at 48 h, and 183 µm^2^ at 72 h (* *p* ˂ 0.05). These results demonstrated the effect of AB1 at a final concentration of 1.2 µg/mL on increasing the cell size of MCF10A cells.

The effect of AB1 at a final concentration of 1.2 and 2.3 µg/mL on the different stages of the cell cycle in the MCF7 cells (Figure 5A) and MCF10A cells (Figure 5B) was evaluated using BD cell cycle/DNA kit and the DNA flow cytometry analysis technique. The results of this experiment showed that, in the untreated control MCF7 cells, 57% of the cells were in the G1 phase, 11.98% of the cells were in the G2 phase, and 30.88% of the cells were in the S1 phase. In the case of the AB1-treated MCF7 cells, the result showed that 74.46%, 8.28%, and 17.26% of the cells were in the G1, G2, and S1 phases, respectively. In the untreated MCF10A cells, 64.29% of the cells were in the G1 phase, 12.29% of the cells were in the G2 phase, and 23.43% of the cells were in the S1 phase. In the AB1-treated MCF10A cells, the result showed that 69.20%, 15.49%, and 15.30% of the cells were in the G1, G2, and S1 phases, respectively. These results imply that AB1 at a final concentration of 1.2 µg/mL on the MCF7 and 2.3 µg/mL on the MCF10A cells caused cell-cycle arrest at the G1 phase.

The effect of AB1 at a final concentration of 1.2 µg/mL on the migration of the MCF7 (Figure 6A and Figure 7A) and at the final concentration of 2.3 µg/mL on the MCF10A (Figure 6B and Figure 7B) cells was carried out to investigate the capability of AB1 in inhibiting or stimulating cell migration. The initial average size of the wound was 989 µm units (* *p* ˂ 0.05). After 20 h of incubation, the wound size of the MCF7 cells reduced to 191 µm, and the control cells reduced to 603 µm (* *p* ˂ 0.05). The total distance of wound closure after AB1 treatment was 798 µm and the control was 386 µm (* *p* ˂ 0.05). The wound closure per hour (µm/h) was calculated and was found to be 39.9 µm with AB1, and 19.3 µm for the untreated cells (* *p* ˂ 0.05). With MCF10A, the initial average size of the wound was 965 µm (* *p* ˂ 0.05) after 20 h of incubation, the wound size in the cells treated with AB1 reduced to 52 µm, and the untreated cells’ wound size reduced to 854 µm (* *p* ˂ 0.05). The total distance of wound closure after AB1 treatment was 913 µm, and the control was 52 µm (* *p* ˂ 0.05). The wound closure per hour (µm/h) was calculated, and it was found to be 45.65 µm with AB1 and 2.6 µm for the untreated cells (* *p* ˂ 0.05).

### 2.4. Gene Expression Levels of Different Tumour-Related Cell Signaling Pathways

The effect of AB1 on tumor-related gene expression in MCF7 and MCF10A was evaluated using RT-qPCR where relative ΔΔCT analysis was implemented. The gene expression levels of tumor suppression genes (*BRCA1*, *BRCA2*, and *p53)*, oncogenes (*HER1*, *HER2*, and *cMyc*), cell cycle genes (*CCND1*, *WNT3A*, *MAPK1*, and *MAPK3*), and apoptosis genes (*BCL2*, *MCL1*, *DAPK1*, *Casp8*, and *Casp9*) were quantified 48-h post treatment; the results are demonstrated in Figure 8 and Figure 9.

### 2.5. Gene Expression Levels of Different Cell Signaling Pathways following AB1 Treatment in MCF7 Cells

#### 2.5.1. AB1 Increased the Gene Expressions of *BRCA2* and *p53* but Reduced the Expression of *BRCA1* in MCF7 Cells

Treatment of AB1 with MCF7 cells showed a significant induction in the gene expression level of *BRCA1* (0.22/*p* = 0.05) with a three-fold increase in the expression of *BRCA2* (3.39 ± 0.18/*p* = 0.015) and eight-fold increase in *p53* (8.48/*p* = 0.001) compared to the control of each gene (Figure 8A).

#### 2.5.2. AB1 Reduced the Gene Expressions of *HER1* and *HER2* but Increased the Expression of *cMyc* in MCF7 Cells

Treatment of AB1 with MCF7 cells showed a significant reduction in the expression of *HER1* and *HER2* by 34% (0.66/*p* = 0.0014) and 79% (0.21/*p* = 0.0001), respectively. On the other hand, treatment of AB1 significantly increased the expression of *cMyc* by 30% (1.3/*p* = 0.001) compared to the control of each gene (Figure 8B).

#### 2.5.3. AB1 Increased the Gene Expressions of *WNT3A* and *MAPK1* but Reduced the Expressions of *CCND1* and *MAPK3* in MCF7 Cells

Treatment of AB1 with MCF7 cells significantly increased the gene expression of *WNT3A* by 10% (1.11/*p* = 0.0423) and *MAPK1* by 39% (1.39/*p* = 0.0232). On the other hand, it significantly reduced the expression levels of *CCND1* by 17% (0.83/*p* = 0.0283) and *MAPK3* by 84% (0.16/*p* = 0.0006) compared to the control of each gene (Figure 8C).

#### 2.5.4. AB1 Increased the Gene Expressions of *BCL2* and *DAPK1* but Reduced the Expressions of *MCL1* and *Casp9* in MCF7 Cells

Treatment of AB1 with MCF7 cells increased the gene expression of *BCL2* by 3.76-fold (4.76/*p* = 0.0451), *DAPK1* by 5.84-fold (6.84/*p* = 0.0018), and *Casp9* by only 65% (1.65/*p* = 0.027) while it significantly reduced the gene expression of *MCL1* 32% (0.68/*p* = 0.01) and *Casp8* by 23% (0.77/*p* = 0.0329) as shown in Figure 8D.

### 2.6. Gene Expression Levels of Different Cell Signaling Pathways following AB1 Treatment in MCF10A Cells

#### 2.6.1. AB1 Reduced the Gene Expressions of *BRCA1*, *BRCA2*, and *p53* in MCF10A Cells

AB1 resulted in a highly significant reduction in the gene expression levels of *BRCA1* (0.25/*p* = 0.0069) by 75%, *BRCA2* (0.061/*p* = 0.0016) by 99.939%, and *p53* (0.055/*p* = 0.0019) by 99.945% compared to the control of each gene as represented in Figure 9A.

#### 2.6.2. AB1 Reduced the Gene Expressions of *HER1*, *HER2*, and *cMyc* in MCF10A Cells

Treatment of AB1 with MCF10A cells reduced the expressions of *HER1* by 73% (0.27/*p* = 0.009), *HER2* by 40% (0.58/*p* = 0.001), and *cMyc* by 70% (0.29/*p* = 0.0012) as compared to the control of respective genes (Figure 9B).

#### 2.6.3. AB1 Reduced the Gene Expressions of *CCND1* and *MAPK3* but Increased the Expression of *MAPK1* in MCF10A Cells

AB1 treated to MCF10A cells reduced the expression of *MAPK3* by 99.91% (0.084/*p* = 0.002) and of *CCND1* by 59% (0.39/*p* = 0.007). On the other hand, AB1 increased the expression of *MAPK1* (16.16/*p* = 0.0038) by 15.16-fold; however, the gene expression of *WNT3A* (1.0/*p* = 0.3194) remained unchanged compared to the control of each gene (Figure 9C).

#### 2.6.4. AB1 Increased the Gene Expressions of *BCL2*, *MCL1*, *DAPK1*, *Casp8*, and *Casp9* in MCF10A Cells

When MCF10A cells were treated with AB1, the gene expression levels of all the apoptosis genes investigated in this study increased, as *BCL2* increased by 101.18-fold (102.18/*p* = 0.0006), *MCL1* (2.34/*p* = 0.013) by 1.3-fold, *DAPK1* (2.97/*p* = 0.092) by 1.9-fold, *Casp8* (3.53/*p* = 0.05) by 2.5-fold, and *Casp9* (13.65/*p* = 0.05) by 12.5-fold compared to the control of each gene (Figure 9D).

## 3. Discussion

Mycotoxins are a group of chemical compounds produced by many fungal species, such as *Aspergillus* and *Penicillium,* which grow on food commodities and produce these toxins as secondary metabolites [1,19]. Several mycotoxins have been discovered to date, but among all of them, Aflatoxin B1 has been characterized as the strongest carcinogenic mycotoxin [20,21,22]. Due to mycotoxins’ chemical compositions, they have been reported to be heat-stable and impossible to completely neutralize during food processing procedures such as sterilization, and their contamination in food will eventually reach the human biological system [23,24]. The effect of AB1 in inducing tumors has been reported in several cancer cell types [12,20], but the effect of AB1 has not been investigated in the breast cancer pathway. The ubiquitous nature of AB1 makes it a toxic metabolite that can cause serious public health concern and its contamination was proven to be hepatotoxic, resulting in liver diseases [25]. In addition, AB1 is believed to have a role in causing hepatic and extrahepatic carcinogenesis in humans by causing single and double DNA breaks [26]. Still, no investigation has been reported to date to link AB1 toxicity and its carcinogenic effect on breast cancer cells. In this study, we attempted to evaluate the effect of AB1 on MCF7 and MCF10A cell lines to investigate its potential to promote tumorigenicity and increase the risk of developing breast cancer.

The first impact of AB1 on MCF7 and MCF10A cells was examined with regards to its ability to cause cellular toxicity, and the results indicated that AB1 decreased cell viability in a concentration-dependent manner, confirming AB1 toxicity towards both cell lines. High concentrations of AB1 [27,28] have a high growth and proliferation inhibition activity in a concentration-dependent manner, confirming AB1 toxicity towards the cells used in this study. AB1 was proven to be carcinogenic in different tissues by means of inducing double-strand breaks in the genome [1,12,29,30], resulting in reducing cellular viability, so it might be plausible that AB1 reduced cell viability by causing DNA damage, which needs further investigation.

The second influence of AB1 on MCF7 and MCF10A was determined by its ability to initiate apoptosis. Our results revealed the ability of AB1 to induce a high apoptosis rate as the concentration of the toxin increased. These results were in agreement with the work reported by Meki et al., which showed AB1′s ability to initiate apoptosis at a high concentration [31]. Meki’s study revealed the influence of AB1 in inducing apoptosis by means of inducing breaks in the genomic DNA strands in rat liver cells [31]. Based on the apoptosis results obtained in this study, it is tempting to speculate the ability of AB1 in inducing apoptosis through DNA double strand breaks, but a DNA fragmentation assay needs to be conducted in the future to confirm this mode of cell death. The third impact of AB1 on the breast cancer cells was investigated through their effect on ROS levels to identify their involvement in either redox signaling or cellular stress. Our results demonstrated reduced levels of ROS by AB1 in breast cancer cells, which contradicts the results of previous studies, where mycotoxins were proven to increase the levels of ROS in vitro due to the production of DNA double strand breakage and increasing cellular stress [32,33]. Besides their ability to cause DNA double strand breaks in the genome and increase cellular stress [34,35], in another study, AB1 was capable of activating Nrf2-dependent signaling, a pathway that results in the production of scavengers that reduce ROS levels in cells [36]. Additionally, low levels of ROS will initiate different cellular pathways [37], which we observed from the gene expression results of the current study. We anticipated the role of AB1 in reducing ROS levels, which subsequently results in initiating redox signaling, affecting the proliferation and growth of treated cells [38]. In addition, low ROS levels have been proven to interfere with the progression of the cell cycle from the G1 to the S phase as cells need ROS at high levels for proliferation [38]. The results of cell cycle progression in this study showed G1 arrest, which supports the involvement of low ROS levels in interfering with cell cycle progression as reported previously [38]. Thus, the role of reduced ROS levels seems to correlate with the cell-cycle arrest at the G1 phase induced by AB1 in both MCF7 and MCF10A cells.

To further investigate the effect of AB1 on the phenotypic characterization of the cells in this study, we observed the changes in cell morphology, cell cycle progression, and cell migration rates following treatment of AB1. Our results demonstrated cell-cycle arrest at the G1 phase in both cells. Cell-cycle arrest at the G1 phase can be related to DNA breaks and the initiation of the redox signaling pathway, which results in epigenetic changes according the work done by Agami and colleagues [39]. Additionally, our results support the work of Huang et al., in which AB1-induced cell-cycle arrest at the S phase due to DNA damage in neuroblastoma cell line [40], and our findings anticipated a similar incident taking place in the MCF7 and MCF10A cells. Cell-cycle arrest is usually accompanied by an increase in cell size, and for that reason, we conducted a cell morphology assay to support our findings. An increase in the cell size was demonstrated, conforming the role of AB1 in altering the cell cycle and causing cell-cycle arrest at the G1 phase [41,42]. Cell arrest at the G1 phase of the cell cycle [43] leads to an increase in cell size, which was demonstrated at different time points post-treatment with AB1. Finally, to further assist the effect of AB1 on MCF7 and MCF10A phenotypic characters, we investigated the migration rates post-treatment. Our finding revealed that the cells were arrested at the G1 phase, but despite that, both cell lines showed high rates of migration post-treatment. According to Bonneton et al., at G1-phase cell-cycle arrest, the NBT-II rat bladder carcinoma cell line showed an increase in cell migration in response to fibroblast growth factor 1 (FGF-1) stimulation [44], and we predict the same mechanism might take place in MCF7 and MCF10A cells in response to AB1 treatment. Additionally, cultured smooth muscle cells demonstrated an increase in migration rates during G1 cell-cycle arrest in response to the stimulation of platelet-derived growth factor B-chain homodimer [45], and we hypothesized that AB1 had a stimulating role in the MCF7 and MCF10A cells, which increased migration rates during cell-cycle arrest at the G1 phase. Both MCF7 [46] and MCF10A cells [47] exhibit chemotactic behavior in response to chemical stimuli, and we can conclude that AB1 exhibited a chemotaxis effect on both these cells resulting in increased migration rates. Finally, the expression level of the *bcl2* gene is critical for the migration and invasion of cancer cells [48]. The low expression levels of BCL2 were reported to inhibit cell migration and proliferation, indicating its important role during cancer cell migration [49]. The high gene expression levels of BCL2 in this study could be a possible reason for the observed fast migration rates despite the cell-cycle arrest induced by AB1 in breast cancer cells.

To further support our findings, we conducted a gene expression analysis to genes controlling the initiation of apoptosis, progression of the cell cycle, and the invasiveness of cancer cells.

### 3.1. Gene Expression of Tumor-Suppressing Genes Indicates Damage to the MCF7 and MCF10A Genome

As the role of AB1 in inducing DNA damage has been proven in previous studies [27], these results suggested that AB1 could be involved in inducing damage in the cellular genome of MCF7 cells, which is required for the activation of *p53* to initiate cell-cycle arrest and to recover from the DNA damage [50]. In addition, the work conducted by Lihua Y. et al. demonstrated the formation of a complex between *BRCA2* and *p53* in vivo, in which the increase in the expression of one gene leads to an increase in expression in the other gene, which explains the increase in *BRCA2* expression [51]. AB1 was proven to downregulate the expression of *BRCA1,* and at the same time, *BRCA1* interacts with many tumor-suppressing genes such as *BRCA2* and *p53*. In the events of *BRCA1* downregulation, *BRCA2* and *p53* will be activated to compensate for the loss in *BRCA1* function [52]. Furthermore, the results of the tumor-suppressing genes showed that when the MCF10A cells were treated with AB1, there was a reduction in the gene expression of *BRCA1*, *BRCA2,* and *p53,* and the decrease in *BRCA1* expression suggests that AB1 affected *BRCA1* expression and activity, which in turn affected the repair mechanism in the MCF10A cells [53]. The reduced expression of *BRCA2* also suggests the interference of AB1 in DNA repair by reducing *BRCA2* expression, which could lead to compromising DNA stability [54]. The reduction in *p53* suggests that apoptosis was not regulated [55,56], and cells were in cell-cycle arrest at the G1 stage. Additionally, the effect of AB1 in inducing a mutation in *p53* was confirmed in human hepatocytes cells [12], which might be a possible effect of AB1 in breast cancer cells.

### 3.2. Gene Expression of Oncogenes Indicates an Increase in MCF7 and MCF10A Invasiveness Post-Treatment with AB1 through Interfering with Proliferation Activities

The reduced gene expression of *HER1* suggests that cell proliferation was altered [57], which explains the cell arrest at the G1 phase. In addition, this result suggests that AB1 might impose a change on the genome of MCF7 that affects *HER1,* possibly interfering with the encoding protein EGFR and eventually interfering with binding to its ligands EGR transforming growth factor, which might prevent the cell from proliferating [58]. The low expression of *cMyc* explains the low apoptosis rate [59], which was associated with high rates of viable cells [60]. The increase in *HER2* was an expected outcome, since the overexpression of *HER2* was reported in many cancers [61], and since AB1 was reported to be carcinogenic in many cell types [62], it is suggested that AB1 has a direct effect on that gene. AB1 significantly reduced the expression of oncogene genes (*HER1*, *cMyc,* and *HER2*) in the MCF10A cells. The reduced expression of *HER1* suggests that AB1 might interfere with the binding of EGFR with EGF preventing its activation, and hence the dimerization and tyrosine autophosphorylation would not take place and the cell proliferation would stop [57], which supports the cell-cycle arrest at the G1 phase. The reduced expression of *HER2* suggests that the binding of *EGFR2* to a ligand-bound epidermal growth factor receptor was interfered with, which would result in the interference of cell proliferation and cell cycle progression [63]. The reduced expression of *cMyc* suggests that apoptosis was interfered with and stopped [59] due to the action of AB1, which explains the high rates of viability and this also suggests the carcinogenic effect of AB1 and the possibility of inducing tumorigenicity in MCF10A cells [12,29].

### 3.3. AB1 Altered Cell Cycle Progression in MCF7 and MCF10A Cells

Previous work argued the importance of cell-cycle arrest for gaining invasiveness activities and a cell-cycle arrest at G1 is of great importance for preparing cells to exhibit invasive activities [64]. An increase in *WNT3A* gene expression could lead to altering both the cell cycle and the developmental process [65]. In addition, the overexpression of *WNT3A* has implications in the process of oncogenesis, which suggests AB1 involvement in promoting tumorgenicity in MCF7 cells [65]. An increase in *MAPK1* expression leads to differentiation, proliferation, and developmental disturbance in MCF7 cells as demonstrated by G1 arrest and low cell death rates [66]. A decrease in *CCND1* expression confirms the cell-cycle altering and interfering effects on the G1/S transition in the cell cycle, which can explain the contribution of AB1 to tumorigenicity in the MCF7 cells [67,68]. When the expression of *MAPK3* was decreased, the cell cycle progression and proliferation of MCF7 cells was altered, which indicates the effect of AB1 in cell-cycle arrest and the prevention of proliferation [69]. In the MCF10A cells, *MAPK3* and *CCND1* expressions were reduced and *MAPK1* expression increased with unchanged *WNT3A* expression. The reduced expression of *MAPK3* suggests that cell proliferation was altered upon treatment with AB1, where *ERKs* encoded by *MAPK3* were prevented from mediating the process of cell differentiation, cell cycle progression, and proliferation [65]. The decreased expression in *CCND1* leads to the mitigation of developmental process [67,68], which explains the cell-cycle arrest at the G1 phase. In a study conducted by Chen et al., the authors explained how an increased *MAPK1* expression caused differentiation, proliferation, and development interruption and cell-cycle arrest [70] leading to prostate cancer. We could speculate that the same pathway might be functional in MCF10A cells based on the elevated expression levels of this gene. Additionally, an increase in the *MAPK1* gene was related to cell-cycle arrest at the G1 phase, which promoted tumorigenesis [71]. This suggests the plausible effect of AB1 in activating a cancer pathway through the overexpression of *MAPK1* in MCF10A cells.

### 3.4. AB1 Promotes Evading Apoptosis, Increasing Cell Viability and Invasiveness

Many cancer cells are known to adopt different pathways to evade apoptosis in order to continue their survival, which will promote cancer cell invasiveness and tumorgenicity [72]. Overexpression of *BCL2* prevents MCF7 cells from initiating apoptosis [73,74]; as a result, there will be an increase in cell size of MCF7 cells without division nor death. In addition, overexpression of *BCL2* has been reported to cause follicular lymphoma [48,75], which can be implicated in MCF7 cells, raising the possibility of increased tumorigenesis and invasiveness in MCF7 cells upon exposure to AB1. The increase in *DAPK1* expression would cause a decrease in cell-cycle progression, forcing the cells to arrest at the G1 phase without undergoing apoptosis [76,77]. Although our results showed an increase in *Casp9* expression, which should lead to apoptosis, the observation of high cell viability without apoptosis suggests that AB1 induced a mutation in the AKT gene, which in turn affected the production of kinase Akt required for the phosphorylation of pro-caspase-9 to initiate apoptosis [78]. Another explanation of high survival rates is the reduction in *MCL1* expression, which will interfere with apoptosis initiation [79]. The decrease in *Casp8* suggests that cell proliferation can be altered [80] and apoptosis rates can be decreased as reported previously [81]. AB1 induced an increase in the expression of apoptosis-related genes such as *BCL2*, *DAPK1*, *Casp8,* and *Casp9* in the MCF10A cells. The increase in the expression of *BCL2* suggests that the apoptosis of the MCF10A cells was interrupted [73,74]. The increase in *DAPK1* expression suggests that the initiation of apoptosis was interfered with due to the alteration in the production of the positive mediator called death-associated protein kinase 1 [76,77], which explains the high cell viability rates. In addition, overexpression of *DAPK1* was reported in many cancers [82,83], which suggests the possibility of AB1 inducing changes to the genome of MCF10A cells, inducing cancer. The marginal increase in *Casp8* and *Casp9* expression suggests that AB1 might interfere with the initiation of apoptosis, resulting in the low apoptosis rates when the MCF10A cells were treated with AB1 [78].

## 4. Materials and Methods

### 4.1. Cell Lines, Culture Media, Seeding Density, and Toxin Concentration

Two cell lines were used in this study, namely Michigan Cancer Foundation-7 (MCF7) and Michigan Cancer Foundation-10A [84,85], which were purchased from American Type Culture Collection (ATCC), USA. A complete growth medium for MCF7 was prepared by adding 50 mL fetal bovine serum (FBS) and 5 mL antibiotic (penicillin/streptomycin) into 500 mL DMEM [86]. For MCF10A cells, the complete growth media was prepared by adding 25 mL horse serum, 100 µL epidermal growth factor (EGF), Hydrocortisone 250 µL, human recombinant insulin 500 µL, and 5 mL antibiotic (penicillin/streptomycin) to 500 mL DMEM media. Both media were stored at 4 °C until further use. Cells were seeded in 96 wells, 6 wells, 25 or 75 cm^2^ cell culture flasks containing complete DMEM media and incubated in a humidified incubator at 37 °C with 5% CO_2_. The final seeding density for both cells was 2.1 × 10^6^ in T75 flasks, 0.7 × 10^6^ in T25 flasks, 0.3 × 10^5^ in 6-well plates, 0.1 × 10^6^ in 12-well plates, 0.05 × 10^6^ in 24-well plates, and 10,000 cells in 96-well plates. The final concentration used to treat MCF7 was 1.2 µg/mL and 2.3 µg/mL for MCF10A. AB1 was purchased from Enzo, USA.

### 4.2. Measurement of Cytotoxicity, Cell Viability and Apoptosis

A cytotoxicity assay was carried out using cell proliferation kit Tetrazolium chloride (2,3-Bis-(2-Methoxy-4-Nitro-5-Sulfophenyl)-2H-Tetrazolium-5-Carboxanilide (XTT) manufactured by ROCHE, Germany [87]. In brief, both cells were seeded in 96-well plates and incubated at 37^°^ C with 5% CO_2_ until they reached 3.2 × 10^4^ confluency. Once confluent, cells were treated with AB1, and the cells were incubated for 48 h. XTT reagent was freshly prepared and added to the cells in the culture plates [88]. Finally, cells were incubated at 37 °C with 5% CO_2_ for 4–6 h. Results were viewed using Thermo Fisher Scientific MULTISKAN SPECTRUM (San Antonio, TX, USA) at 450–500 nm. The calculation of cell viability was carried out using excel and cell viability calculation formula [87,89]. Apoptosis level measurement was carried out using an ANNEXIN V-FITC Apoptosis Detection Kit [90]. In brief, cells were seeded in 6-well plates using growth media accordingly and allowed to grow to reach a confluence of 1.2 × 10^6^. Once the cells were confluent, the cells were treated with AB1, and the cells were incubated for 48 h at 37 °C with 5% CO_2_. Positive control was prepared by treating the cells with puromycin at a concentration of 30 µg/mL to induce apoptosis, and for the negative control, the cells were allowed to grow without treatment. Cells were detached, collected in 10 mL tubes, and were centrifuged at 300× *g* for 5 min. To the cell pellet, binding buffer was added with 5 µL of annexin V-FITC followed by 10 min incubation in the dark. An additional 5 µL of propidium iodide was added followed by an incubation period of 10 min in the dark followed by a wash with supplement buffer and centrifugation. Finally, 500 µL of fresh binding buffer was added to the cells and samples were analyzed using BD FACSCanto^TM^ II (Piscataway, NJ, USA) flow cytometry for the detection of apoptosis occurrence in the cells post-treatment.

### 4.3. Measurement of Reactive Oxygen Species (ROS) Levels

This experiment was carried out using ROS Detection Cell-Based Assay Kit (DHE) Cayman Chemical, Ann Arbor, MI, USA [91], where cells were seeded in 6-well plates and allowed to reach a confluency of 1.2 × 10^6^. Once cells were confluent, fresh media containing the toxin was added and cells were incubated for 48 h at 37 °C with 5% CO_2_. Cells were detached and collected in 10 mL tubes, centrifuged, and washed with 1 mL of supplement buffer. To the cell pellet, fresh 100 µL binding buffer supplemented with 0.1 µL of DHE probe was added, and the cells were incubated in the dark for 30 min. Finally, the cells were washed twice with supplement buffer, and a fresh 500 µL of binding buffer was added and the sample was analyzed for detecting the ROS levels in the cells using BD FACSCanto^TM^ II flow cytometry.

### 4.4. Measurement Cell Morphology, Cell Migration Measurement, and Cell Cycle Progression

For cell morphology, cells were seeded in 6-well plates and allowed to reach confluency. Once the cells were confluent, fresh media containing AB1 was added, and the cells were incubated. Changes were observed and documented using an OLYMPUS IX51 inverted-light microscope (10×) at 0, 24, 48, and 72 h post-treatment. Cell morphology changes such as granularity, cytoplasmic vacuolation, and changes in cells size were investigated [92]. Pictures were analyzed using the Image J software. For cell migration, cells were seeded in the corresponding density in 12-well plates and incubated under the appropriate conditions until reaching confluency. When the cells were confluent, a vertical wound was created on the surface of each well using a 200 µL tip and the cells were washed with 1X PBS twice. A fresh media containing the toxin was added and cells were allowed to grow, and their migration was monitored and documented. Pictures of the wound size and cell migration were taken using an OLYMPUS IX51 inverted-light microscope with 10× magnification at 6, 12, 18, 20, and 24 h [93,94]. Migration rates were analyzed using Motic Images Plus 3.0 ML. A cell cycle assay was carried out using BD cell cycle/DNA kit by BD, USA [95]. MCF7 and MCF10A cells were seeded in 6-well plates using their respective growth media for each cell line and allowed to grow to reach confluency. Once the cells were confluent, fresh media containing AB1 was added to cells and the cells were incubated for 48 h at 37 °C with 5% CO_2_. After incubation, cells were detached and collected in 10 mL tubes and then centrifuged, and solutions A, B, and C were added according to the manufacturer manual. The final sample contained the cells, and solutions A, B, and C were analyzed using BD FACS Canto^TM^ II flow cytometry for the detection of the genomic DNA content, the different cell cycle stages, and cell arrest occurrence [96].

### 4.5. Measurement of mRNA and Gene Expression Levels

Gene expressions of the different genes in this study were quantified using SensiFAST™ SYBR ^®^ Hi-ROX Kit (Bioline, Little Clacton, Essex, UK) [97]. Firstly, MCF7 cells were seeded in 6-well plates and incubated under the appropriate conditions and allowed to reach confluency. Once the cells were confluent, fresh media containing the toxins were added to the cells and the cells were incubated for 48 h. Next, RNA extraction was performed using Tri-Reagent manufactured by Molecular Research Centre, Inc., Cincinnati, OH, USA, and the manufacturer protocol was followed. The concentration of RNA was measured using Thermo-Fisher Scientific NanoDrop 2000c. cDNA synthesis was conducted using a Tetro cDNA Synthesis Kit (Bioline, UK). The RT-qPCR reaction was prepared on a 0.2 mL low-profile strip with optical clear flat caps (LABCON, Petaluma, CA, USA) and the RT-qPCR reaction was set according to the manufacturer manual. The primers used in this study to investigate the mRNA levels of the targeted genes are listed in Table 1.

### 4.6. Statistical Analysis

All statistical analyses in this study were conducted using GraphPad Prism 8.0 (San Diego, CA, USA). Numerical data values in this study were expressed in the form of mean ± standard deviation where *n* = 3, and statistical significance between the groups was determined by a two-tailed *t*-test and significant results are represented as * *p* ≤ 0.05, ** *p* ≤ 0.01, and *** *p* ≤ 0.001.

## 5. Conclusions

This study reported, for the first time, the effect of AB1 imposed on the phenotypic and genotypic characteristics of MCF7 and MCF10A cells to identify the potential pathways of tumorigenicity of AB1 towards breast cancer. AB1 induced cytotoxicity and apoptosis in both the tested cell lines. AB1 treatment caused hypertrophy with increased cell size, increased migration rates, and altered the gene expression of tumor-related genes, including oncogenes, tumor-suppressing genes, cell cycle genes, and apoptosis genes. This study provides insights on the different mechanisms and physical changes taking place in the normal and breast cancer cell lines post treatment with AB1 and highlights the potential effects of this toxin in inducing tumor-related gene expression, which potentially could increase the risk of breast cancer. These results significantly impact our understanding of the effects of AB1 on gene expression as well as cellular functions and help drive future advancements in breast cancer research. Nevertheless, future studies are needed to better understand the potential role of AB1 in increasing the risk of breast cancer.

## Figures and Tables

**Figure 1 ijms-23-11856-f001:**
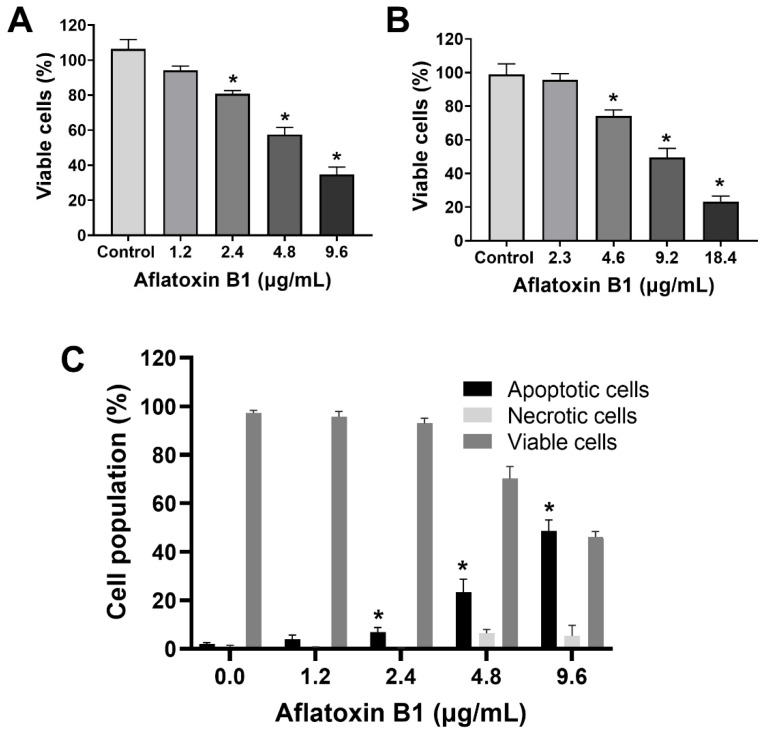
Cell viability and cell death analyses of MCF7 (**A**,**C**) and MCF10A (**B**,**D**) cells after treatment with aflatoxin B1. * *p <* 0.05 compared to control.

**Figure 2 ijms-23-11856-f002:**
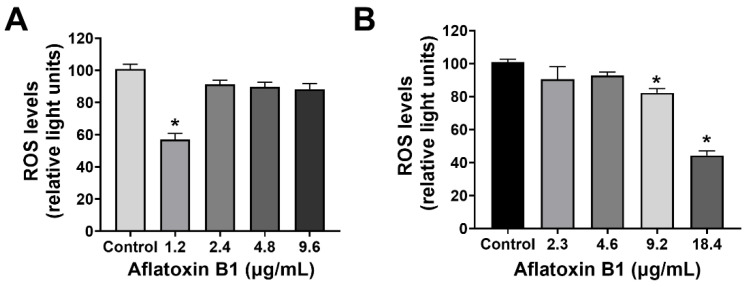
ROS levels of MCF7 (**A**) and MCF10A (**B**) cells after treatment with aflatoxin B1. * *p <* 0.05 compared to control.

**Figure 3 ijms-23-11856-f003:**
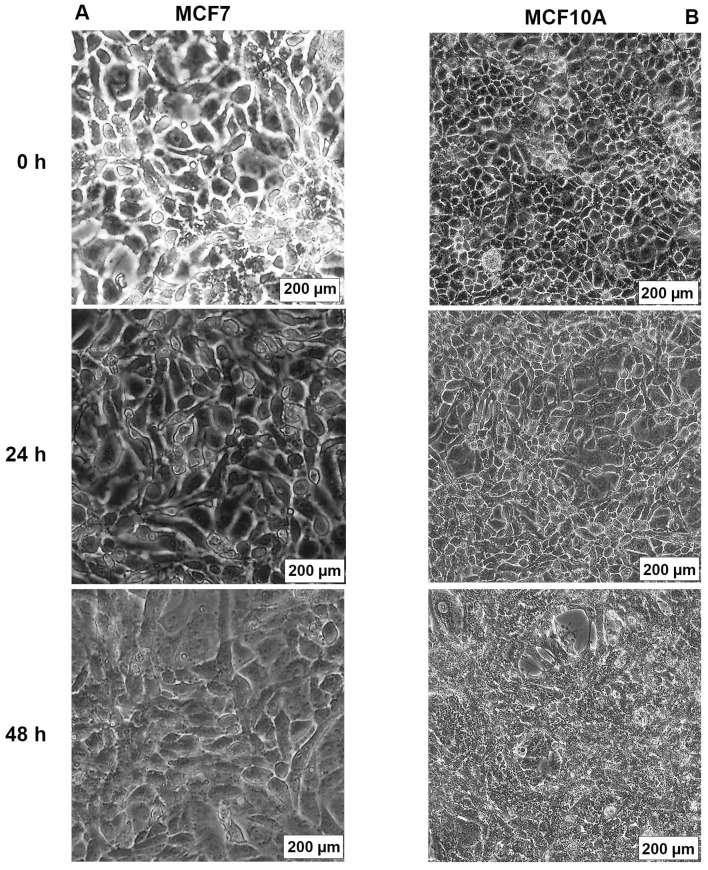
Morphological changes in MCF7 (**A**) and MCF10A (**B**) cells after treatment with aflatoxin B1 from 0–72 h.

**Figure 4 ijms-23-11856-f004:**
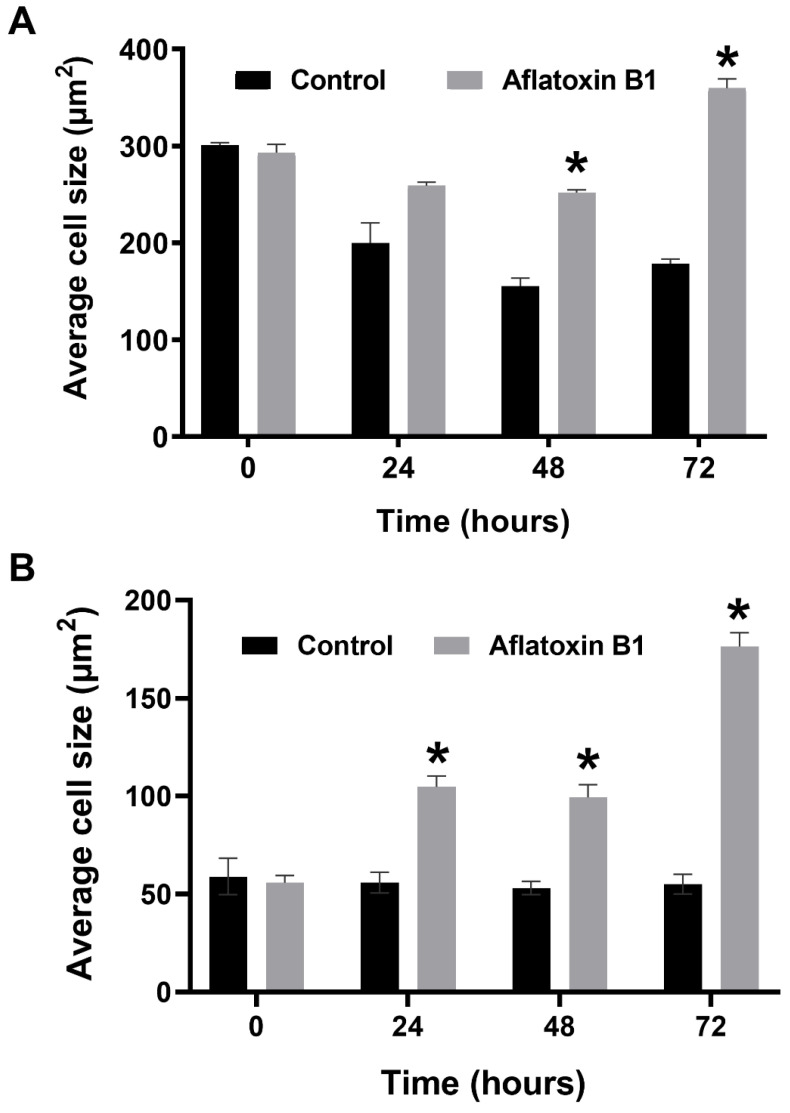
Cell size of MCF7 (**A**) and MCF10A (**B**) cells after treatment with aflatoxin B1. * *p* < 0.05 as compared to control.

**Figure 5 ijms-23-11856-f005:**
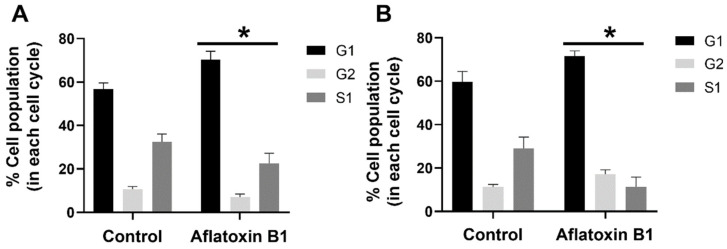
Percentage of MCF7 (**A**) and MCF10A (**B**) cells at different stages of the cell cycle after treatment with aflatoxin B1. * *p <* 0.05 as compared to control.

**Figure 6 ijms-23-11856-f006:**
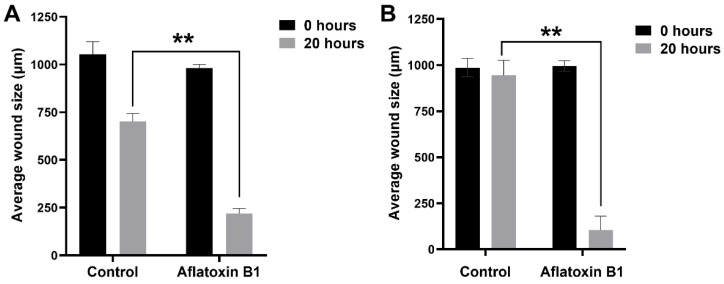
The effects of aflatoxin B1 on cell migration of MCF7 (**A**) and MCF10A (**B**) cells. ** *p* < 0.01 as compared to control.

**Figure 7 ijms-23-11856-f007:**
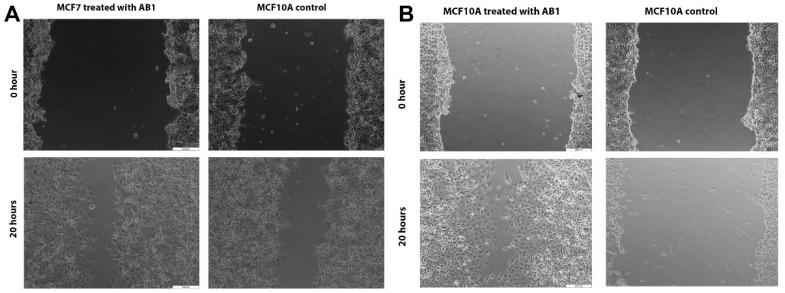
Microscopic images of the effects of aflatoxin B1 on cell migration of MCF7 (**A**) and MCF10A (**B**) cells. The images were obtained at 200 µm scale magnification.

**Figure 8 ijms-23-11856-f008:**
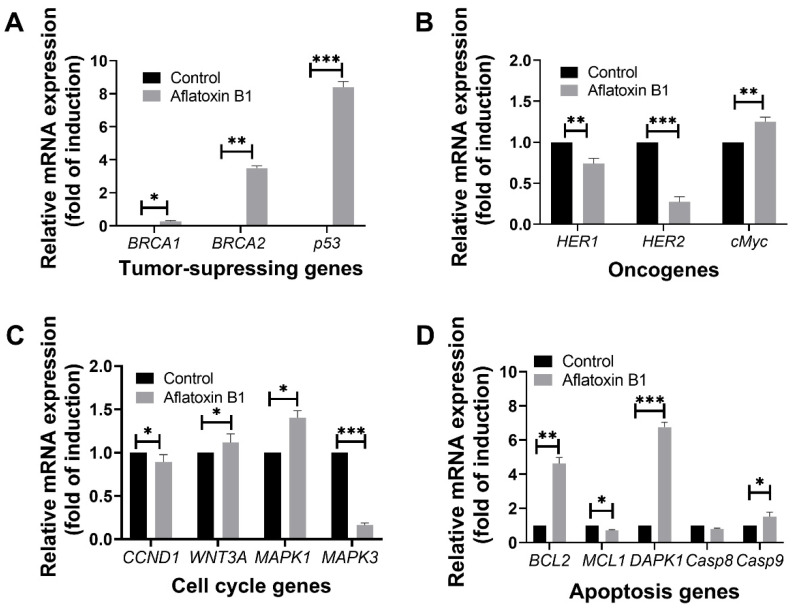
The effects of aflatoxin B1 on tumor-related gene expressions like tumor suppressing (**A**), onco- (**B**), cell cycle (**C**), and apoptosis (**D**) genes in MCF7 cells. * *p <* 0.05, ** *p <* 0.01 and *** *p <* 0.001 compared to control.

**Figure 9 ijms-23-11856-f009:**
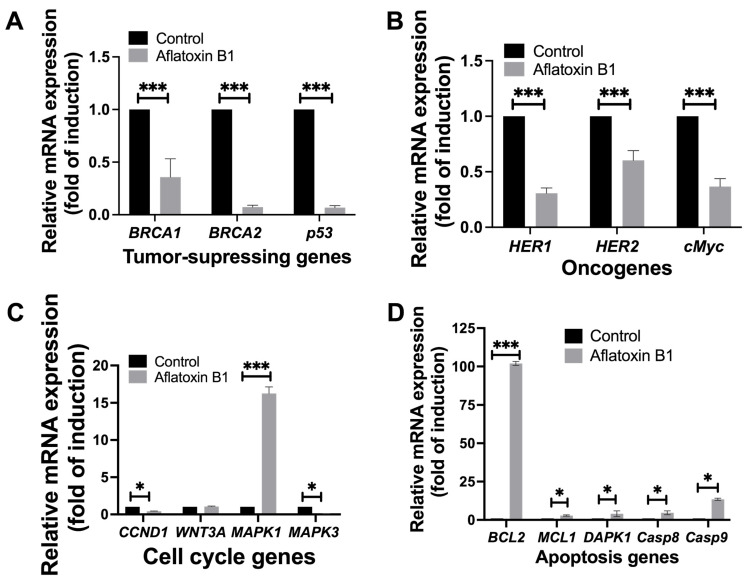
The effects of aflatoxin B1 on tumor-related gene expressions like tumor suppressing (**A**), onco- (**B**), cell cycle (**C**), and apoptosis (**D**) genes in MCF10A cells. * *p <* 0.05 and *** *p <* 0.001 compared to control.

**Table 1 ijms-23-11856-t001:** Primers used in this study.

Gene Name	Primer Sequences
BRCA1	FW-5′-TCGTATTCTGAGAGGCTGCTG-3′ RE-5′-TCTTCAACGCGAAGAGCAGA-3′
BRCA2	FW-5′-GTTGTGAAAAAAACAGGACTTG-3′ RE-5′-CAGTCTTTAGTTGGGGTGGA-3′
p53	FW-5′-AGGCCTTGGAACTCAAGGAT-3′ RE-5′-CCCTTTTTGGACTTCAGGTG-3′
HER1	FW-5′-CAGCGCTACCTTGTCATTCA-3′ RE-5′-TGCACTCAGAGAGCTCAGGA-3′
HER2	FW-5′-AAAGGCCCAAGACTCTCTCC-3′ RE-5′-CAAGTACTCGGGGTTCTCCA-3′
cMyc	FW-5′-TGAGGAGACACCGCCCAC-3′ RE-5′-CAACATCGATTTCTTCCTCATCTTC-3′
BCL2	FW-5′-GAACTGGGGGAGGATTGTGG-3′ RE-5′-GCCGGTTCAGGTACTCAGTC-3′
MCL1	FW-5′-TTCCAGTAAGGAGTCGGGGT-3′ RE-5′-TGGCCAAAAGTCGCCCTC-3′
CCND1	FW-5′-TTCAAATGTGTGCAGAAGGA’3 RE-5′-GGGATGGTCTCCTTCATCTT-3′
WNT3A	FW-5′-GTGTTCCACTGGTGCTGCTA-3′ RE-5′-CCCTGCCTTCAGGTAGGAGT-3′
MAPK1	FW-5′-CAGTTCTTGACCCCTGGTCC-3′ RE-5′-TACATACTGCCGCAGGTCAC-3′
MAPK3	FW-5′-TATGACCACGTGCGCAAGAC-3′ RE-5′-GACATTCTCATGGCGGAAGC-3′
DAPK1	FW-5′-TGGAGAGAGATTGCTCCCAGT-3′ RE-5′-CACAACCGCAAACTGTCCAC-3′
Casp8	FW-5′-CTGGTCTGAAGGCTGGTTGT-3′ RE-5′-CAGGCTCAGGAACTTGAGGG-3′
Casp9	Fw-5′-CAGGCCCCATATGATCGAGG-3′ Re-5′-TCGACAACTTTGCTGCTTGC-3′
Β-Actin	FW-5′-AGAGCTACGAGCTGCCTGAC-3′ RE-5′-AGCACTGTGTTGGCGTACAG-3′

## Data Availability

The data presented in this study are available on request from the corresponding author.

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
