# Peer review of "The Effect of Aflatoxin B1 on Tumor-Related Genes and Phenotypic Characters of MCF7 and MCF10A Cells"

_ijms, 2022, doi:10.3390/ijms231911856_

Round 1
Reviewer 1 Report
In the present manuscript, the authors evaluated the effect of AB1 exposure in breast cancer MCF7 and MCF10A cell lines to evaluate its potential in increasing the risk of breast cancer development.
This paper is interesting and could be a further support to the studies already present and a starting point for further studies. Moreover, the methods used conform to the analysis carried out.
The manuscript can be accepted for publication if the authors are ready to incorporate the following revisions:
Abstract
There seems to be an error in the sentence: “We hypothesized that even at low concentrations, AB1 can cause changes in the expression of important genes involves in four pathways that is cancer, cell cycle, and apoptosis”, please correct.
In abstract and the text, there is no indication of what the acronym OTA means.
Introduction
In the introduction, the authors should indicate the reason why they decide to evaluate the expression of only these genes (because they are involved in different pathways, etc, as indicated in the abstract). Also, they should mention that there are also additional genes involved in such pathways, albeit not analyzed, including PALB2, RAD51, CHEK2.
In this regard, the authors would benefit from reading the following articles, the contents of which could be useful for improving the manuscript:
PMID: 25529982 DOI: 10.3892/or.2014.3685
PMID: 31575382 DOI: 10.3727/096504019X15698362825407
Results
In the initial part of each paragraph of the results, the method used is indicated, for example in section 2.1: “The toxic effect of AB1 on MCF7 cells was examined by using cell proliferation kit (XTT) to determine the percentage of viable cells post treatment. Cells were treated with AB1 in different concentration concentrations (1.2, 2.4, 4.8, and 9.6 μg/mL) and they were incubated for 48 hours”. These parts must be moved to ‘materials and methods’ section and only the results obtained must be indicated here.
Discussion
I believe that the discussion should not be divided into paragraphs.
Author Response
We thank the reviewers for their constructive comments. We have addressed all the comments as per their suggestions and believe the manuscript is now suitable for publication.
Author’s Reply to the Review Report for Reviewer 1
Abstract
There seems to be an error in the sentence: We hypothesized that even at low concentrations, AB1 can cause changes in the expression of important genes involves in four pathways that is cancer, cell cycle, and apoptosis”, please correct.
In abstract and the text, there is no indication of what the acronym OTA means.
All the errors have been corrected.
Introduction
In the introduction, the authors should indicate the reason why they decide to evaluate the expression of only these genes (because they are involved in different pathways, etc, as indicated in the abstract). Also, they should mention that there are also additional genes involved in such pathways, albeit not analyzed, including PALB2, RAD51, CHEK2.
In this regard, the authors would benefit from reading the following articles, the contents of which could be useful for improving the manuscript:
PMID: 25529982 DOI: 10.3892/or.2014.3685
PMID: 31575382 DOI: 10.3727/096504019X15698362825407
As per the reviewer’s suggestion, we have introduced the justification of selecting only a few genes for our study in addition to citing the other articles of importance involving other non-selected genes (as described below)
“In addition, previous work indicates that these gene groups involve in tuning cancer phenotype. So, in this study, selected genes were chosen due to their extensive backgrounds in previous studies. We acknowledge that several other genes involved in these pathways, including PALB2, RAD51 and CHEK2”.
Results
In the initial part of each paragraph of the results, the method used is indicated, for example in section 2.1: “The toxic effect of AB1 on MCF7 cells was examined by using cell proliferation kit (XTT) to determine the percentage of viable cells post treatment. Cells were treated with AB1 in different concentration concentrations (1.2, 2.4, 4.8, and 9.6 μg/mL) and they were incubated for 48 hours”. These parts must be moved to ‘materials and methods’ section and only the results obtained must be indicated here.
The result’s section of the manuscript has now been revised as per the suggestion of the reviewer.
Discussion
I believe that the discussion should not be divided into paragraphs.
As the other reviewer suggests improving the Discussion section highlighting in a simple way the key points of the results obtained, we have included the subheadings with key findings which demanded us to retain the paragraphs in the Discussion section.
Reviewer 2 Report
Aflatoxin B1 is a class 1 carcinogen and it is a well known hepatic toxin and carcinogen. Several studies investigated the role of Aflatoxin B1 as carcinogenic factor for other type of cancer (for example lung and gallbladder cancer) but no evidence exist in the litterature about the research of potential effects of AB1 in increasing the risk of breast cancer. For this reason the topic of this manuscript is of great clinical interest. The materials and method section is clearly explained and make the study reproducibile. The conclusions should be improved highlighting in a simple way the key points of the results obtained. Correction of typing errors is required as well as a careful revision of the English language.
Author Response
We thank the reviewers for their constructive comments. We have addressed all the comments as per their suggestions and believe the manuscript is now suitable for publication.
Author’s Reply to the Review Report for Reviewer 2
Comments and Suggestions for Authors
The conclusions should be improved highlighting in a simple way the key points of the results obtained. Correction of typing errors is required as well as a careful revision of the English language.
A conclusion section has been now included with all the key findings. The manuscript has been carefully revised for English language editing.